# Effect of Organic Substances on Nutrients Recovery by Struvite Electrochemical Precipitation from Synthetic Anaerobically Treated Swine Wastewater

**DOI:** 10.3390/membranes11080594

**Published:** 2021-08-04

**Authors:** Run-Feng Chen, Tao Liu, Hong-Wei Rong, Hai-Tao Zhong, Chun-Hai Wei

**Affiliations:** 1Department of Municipal Engineering, School of Civil Engineering, Guangzhou University, Guangzhou 510006, China; gzdxcrf@163.com (R.-F.C.); liutao@gzhu.edu.cn (T.L.); hwrong@gzhu.edu.cn (H.-W.R.); 2Key Laboratory for Water Quality and Conservation of the Pearl River Delta, Ministry of Education, Guangzhou 510006, China; 3College of Resources and Environment, South China Agricultural University, Guangzhou 510642, China; zhonghaitao815@163.com

**Keywords:** organic substances, nutrients recovery, struvite electrochemical precipitation, swine wastewater, initial pH

## Abstract

Anaerobically treated swine wastewater contains large amounts of orthophosphate phosphorus, ammonium nitrogen and organic substances with potential nutrients recovery via struvite electrochemical precipitation post-treatment. Lab-scale batch experiments were systematically conducted in this study to investigate the effects of initial pH, current density, organic substances upon nutrients removal, and precipitates quality (characterized by X-ray diffraction, scanning electron microscopy and element analysis via acid dissolution method) during the struvite electrochemical precipitation process. The optimal conditions for the initial pH of 7.0 and current density of 4 mA/cm^2^ favoured nutrients removal and precipitates quality (struvite purity of up to 94.2%) in the absence of organic substances. By contrast, a more adverse effect on nutrients removal, morphology and purity of precipitates was found by humic acid than by sodium alginate and bovine albumin in the individual presence of organic substances. Low concentration combination of bovine albumin, sodium alginate, and humic acid showed antagonistic inhibition effects, whereas a high concentration combination showed the accelerating inhibition effects. Initial pH adjustment from 7 to 8 could effectively mitigate the adverse effects on struvite electrochemical precipitation under high concentration combined with organic substances (500 mg/L bovine albumin, 500 mg/L sodium alginate, and 1500 mg/L humic acid); this may help improve struvite electrochemical precipitation technology in practical application for nutrients recovery from anaerobically treated swine wastewater.

## 1. Introduction

Swine production intensifies with the increase in meat demands globally and is one of the fastest growing livestock industries. The massive swine population produces a large quantity of heavily polluted swine wastewater, which ultimately and adversely affects the environment if not properly treated. Therefore, the treatment and management of swine wastewater is a high priority issue in many countries worldwide [1]. Organic matter and nutrients such as nitrogen and phosphorus in swine wastewater have the dual properties of real environmental pollutants and potential energy and resource carriers. It is a winning strategy to convert organics and nutrients in swine wastewater into energy resources and fertilizer as well as produce qualified effluent for safe discharge or reuse via treatment technology with high efficiency and low cost for ecological, environmental protection, and socioeconomic development.

Anaerobic biological treatment technologies have the advantages of a high organic loading rate, methane production as an energy resource, no aeration, and low energy requirements [2], which have become the core process for swine wastewater treatment [3]. Both conventional anaerobic technologies (e.g., anaerobic digestion [4], anaerobic sequencing batch reactor [5], anaerobic baffled reactor [6]); and advanced aerobic technologies, such as an anaerobic membrane bioreactor [7]) have been used. Swine wastewater treatment especially benefits from these technologies’ excellent capacity to separate sludge and water by micro/ultra-filtration membrane instead of gravitational sedimentation; anaerobic membrane bioreactor can also produce effluent free of suspended solids and thus facilitate further treatment. Although it can effectively remove organics from swine wastewater, anaerobic treatment cannot remove nutrients. On the contrary, it even enhances its concentration in effluent mainly due to converting organic nitrogen and phosphorus into inorganic ammonium and orthophosphate [8,9], which requires further treatment. In reality, the anaerobically treated swine wastewater is rich in ammonium nitrogen and orthophosphate phosphorus, indicating a high potential for struvite recovery [10].

Nutrient recovery from wastewater by precipitation in the form of crystalline struvite is an attractive option for sustainable development considering the commercial value of struvite (MgNH_4_PO_4_·6H_2_O) as a slow-release fertilizer. As long as the crystallization conditions are suitable, the struvite can be generated by adding magnesium ions (Mg^2+^) into swine wastewater to remove ammonium nitrogen and orthophosphate phosphorus simultaneously. Struvite precipitation occurs under alkaline conditions according to the reaction as follows: Mg^2+^ + NH_4_^+^ + H_n_PO_4_^3−n^ + 6H_2_O → MgNH_4_PO_4_·6H_2_O + nH^+^, where n equals 0, 1, or 2 based on solution pH [11].

It was previously reported that the substrates’ saturation depending on the concentration, pH, the Mg^2+^:NH_4_^+^:PO_4_^3−^ molar ratio, temperature, mixing energy, and suspended solids, had significant effects on struvite precipitation [11,12,13]. Some inorganic ions in swine wastewater, such as CO_3_^2−^, Ca^2+^ and K^+^, can interfere with the nucleation of crystal and affect the struvite precipitation process, leading to a decreased recovery [14]. In fact, anaerobically treated swine wastewater not only contains large amounts of ammonium nitrogen, orthophosphate phosphorus, and ions but many organic substances, including anaerobic metabolites (such as volatile fatty acids, polysaccharides, proteins, humics). These organic substances will inevitably pose an adverse effect on the crystallization and growth of struvite, and even its crystallization efficiency and quality, limiting the application of crystals as slow-release fertilizers [15].

Currently, only a few studies were conducted to assess the effect of organic substances on the struvite precipitation process with chemical reagent addition. Song et al. [14] found that citric acid had a more significant influence than succinic acid and acetic acid on synthetic wastewater and struvite crystallization. Likewise, Zhou et al. [16] found that humics not only had an inhibition effect on phosphorus removal efficiency but could also negatively affect the crystal form, morphology, and purity of the harvested struvite precipitates, which could be mitigated by increasing the Mg: P ratio and solution pH. Huang et al. [17] reported that sodium alginate and bovine albumin slightly influenced the crystallization of struvite while acetic acid did not. Recently, Muhmood et al. [9] reported that organic substances reduced nutrient recovery only at relatively low pH (8.0–9.0), and the particulate and colloidal organic substances from real high-strength wastewater slowed the precipitation reaction, substantially increased the particle size, and did not significantly influence the purity and crystal structure of struvite.

However, there is little information on the effect of organic substances in anaerobically treated swine wastewater (e.g., polysaccharides, proteins, humics) on nutrients recovery by struvite electrochemical precipitation, which is an attractive alternative method to the traditional process with chemical reagent addition because adding magnesium is accomplished with pH increase during electrolysis [18]. Notably, few studies consider the coexistence of these organic substances during the recovery of nutrients through struvite electrochemical precipitation.

Therefore, to address this knowledge gap, this study was undertaken to explore the effects of representative organic substances (sodium alginate on behalf of polysaccharides, bovine albumin on behalf of proteins and humic acid on behalf of humics) in synthetic anaerobically treated swine wastewater on struvite electrochemical precipitation performance (indicated by nutrient recovery efficiency and precipitate quality). The results will help clarify the mechanism of organics affecting struvite electrochemical precipitation and explore the optimum operational parameters to mitigate the adverse effects of organics.

## 2. Materials and Methods

### 2.1. Wastewater Characterisation

Major inorganic contaminants (average concentrations of PO_4_^3−^-P, NH_4_^+^-N, K^+^, CO_3_^2−^, SO_4_^2−^ were 132, 520, 375, 1435, and 573 mg/L, respectively) were added into the synthetic swine wastewater according to the real swine wastewater reported previously [19]. Other trace elements (e.g., Fe, Zn, Cu, Cr, Co, Ni) with concentrations less than 0.2 mg/L in real swine wastewater were not added to the synthetic wastewater due to their marginal effects on struvite electrochemical precipitation from previous reports [11,12]. Suspended solids were also not added to the synthetic wastewater to simulate the effluent from the anaerobic membrane bioreactor. In order to avoid interference from complex components, stock solutions were prepared to form synthetic swine wastewater by dissolving an analytical reagent grade of KH_2_PO_4_ and NH_4_Cl chemicals in deionized water to obtain the initial orthophosphate phosphorus and ammonium nitrogen concentration of 132 and 520 mg/L, respectively. When investigating the effect of organic substances, the major ions (K^+^, CO_3_^2−^ and SO_4_^2−^) were added to the solution matrix to maximise the reflection of real wastewater properties.

The wastewater treated by an anaerobic process generally contains a certain amount of extracellular polymeric substances generated from microbial metabolism. The extracellular polymeric substances mainly comprise various types of high molecular weight materials, such as polysaccharides, proteins, nucleic acids, and so on [17]. Sodium alginate and bovine albumin, which have properties similar to those of polysaccharides and proteins in the extracellular matrix, respectively, were chosen as the simulated reagents of extracellular polymeric substances in anaerobically treated swine wastewater. Additionally, digested swine slurry often contains a high content of various organic matters, and humic substances are sometimes found to be one of the more important fractions [20]. Thus, humic acid, one of the major fractions in wastewater, was also chosen to simulate the low molecular weight organics in anaerobically treated swine wastewater.

The chemical reagents used in the experiments, including NH_4_Cl, KH_2_PO_4_, KCl, Na_2_CO_3_, Na_2_SO_4_, sodium alginate (SA), bovine serum albumin (BSA), and humic acid (HA), were analytical grade. Deionized water was used in all the experiments. The whole experiment was divided into two parts for the two kinds of synthetic swine wastewater used. Firstly, the synthetic swine wastewater, only containing orthophosphate phosphorus and ammonium nitrogen, was used to investigate the characteristics of struvite electrochemical precipitation under the controlled initial pH and current density, respectively. Afterwards, the synthetic swine wastewater, containing orthophosphate phosphorus, ammonium nitrogen and organic substances, with major ions of K^+^, CO_3_^2−^ and SO_4_^2−^, was used to investigate the effects of organic substances (SA, BSA, HA, and their combinations with different concentrations) on struvite electrochemical precipitation performance. The experiments for individual effects were conducted by preparing different concentrations of BSA (100, 200, 500 mg/L), SA (100, 200, 500 mg/L), and HA (200, 500, 1500 mg/L). The experiments for combined effects were conducted by preparing different combinations of BSA (500 mg/L) + SA (500 mg/L), BSA (500 mg/L) + HA (1500 mg/L), SA (500 mg/L) + HA (1500 mg/L), BSA (100 mg/L) + SA (100 mg/L) + HA (200 mg/L), and BSA (500 mg/L) + SA (500 mg/L) + HA 1500 mg/L), where the last two combinations were aimed at simulating anaerobically treated swine wastewater by conventional and advanced technology (e.g., anaerobic digestion and anaerobic membrane bioreactor), respectively.

### 2.2. Experiment Setup

A schematic diagram of the experimental setup is provided in Figure 1. The electrolysis cell consists of a 1000 mL beaker on a magnetic stirrer, two rectangular plate electrodes with an effective size of 3.6 × 5.0 × 0.2 cm and a DC power supply (RXN-305DM, 0~30 V, 0~5 A, ZHAOXIN). The two electrodes connected in a monopolar mode were dipped in the beaker, and their distance was 2 cm. The magnesium (Mg) anode and the stainless steel cathode were connected to the positive and negative poles of the DC power supply, respectively. The effective surface area (i.e., the immersed part) of both electrodes was 36 cm^2^ (3.6 cm × 5.0 cm × 2 faces). Before each test, the electrode plates were scrubbed with a coarse sandpaper and then rinsed with 10% HCl to remove the surface attachment. There was a gap of 2 cm between the beaker bottom and the electrodes to facilitate continuous stirring (200 rpm).

### 2.3. Batch Experiments

The batch experiments consisted of two parts regarding the synthetic wastewater used. For the first past with synthetic wastewater containing only orthophosphate phosphorus and ammonium nitrogen, the batch tests under a controlled initial pH (6.5, 7.0, 7.5 and 8.0) and current density (2, 4 and 6 mA/cm^2^) were conducted to investigate orthophosphate phosphorus and ammonium nitrogen recovery by struvite electrochemical precipitation process. For the second part with synthetic wastewater containing orthophosphate phosphorus, ammonium nitrogen, major ions, and organics (BSA, SA, HA, and their combinations) simulating anaerobically treated swine wastewater, the batch tests under the initial pH (7.0) and current density (4 mA/cm^2^) were conducted to investigate the effect of organic substances on struvite electrochemical precipitation performance.

For each batch test, wastewater of 500 mL was filled in the beaker, and the initial pH was adjusted to the desired value using 10% HCl and 20% NaOH solution. The treated wastewater quality parameters (orthophosphate phosphorus and ammonium nitrogen) were monitored at different time intervals during the test. Samples of 2 mL in the beaker were collected from the middle supernatant and filtrated through a 0.45 μm membrane syringe filter for measurement. In addition, the pH and conductivity of the wastewater were measured in situ using a dual pH and conductivity meter (HQ40d, Hach). At the end of each experiment, the precipitate in the solution was collected using filter paper and dried at room temperature for 48 h. The structure, morphology, and purity of the harvested precipitate were analysed by X-ray diffraction (XRD), scanning electron microscopy (SEM), and element analysis by acid dissolution method, respectively. All batch tests were conducted at room temperature in duplicate.

### 2.4. Analytical Methods

The concentrations of ammonium nitrogen and orthophosphate phosphorus in the solution were determined by the standard methods [21] of Nessler’s reagent spectrophotometry and molybdenum-antimony anti-spectrophotometry, respectively. The cations contents were detected by an inductively coupled plasma mass spectrometry (ICP-MS) (NexION 300, PerkinElmer, Waltham, MA, USA). The anions contents were detected by ion chromatography (IC) (ICS-2000, Dionex, Sunnyvale, CA, USA). The microscopic morphology of the harvested precipitate was observed by SEM (JSM-7001F, JEOL, Tokyo, Japan). The crystal form of the harvested precipitate was determined by XRD (PW3040/60, PANalytical B.V., Almelo, The Netherlands). The purity of the harvested precipitate was determined by element analysis by the acid dissolution method. Approximately 50 mg of the struvite precipitates were dissolved with a small amount of 0.1 mol/L HCl solution and then diluted to 100 mL with deionized water. Before ammonium nitrogen determination, the pH of the solution was adjusted to 4 using a 1 mol/L NaOH solution [22,23]. Most of the common struvite mineral impurities do not contain nitrogen, i.e., Mg(OH)_2_, MgHPO_4_, Mg_3_(PO_4_)_2_, MgKPO_4_, CaHPO_4_, Ca_5_(PO_4_)_3_OH [13]. Struvite purity was quantified based on the assumption that 1 mol of ammonium is equivalent to 1 mol of struvite. Therefore, the struvite purity can be calculated according to the following equation:struvite purity (%) = (C_ammonium_ × V × M_struvite_)/(m_precipitate_ × M_N_) × 100(1)
where C_ammonium_ is the ammonium nitrogen concentration, mg/L; V is the constant volume, 0.1 L; M_struvite_ is the molar mass of struvite, 245 g/mol; m_precipitate_ is the mass of the precipitate, mg; M_N_ is the molar mass of nitrogen, 14 g/mol. The purity of the pure struvite purchased from Macklin (cas: 13478-16-5) was calculated using the equation at the level of 98.6% (average), which matched well with the labelled content (98.0%) of the pure struvite. Thus, the equation could be reliably used to determine other struvite purity of the precipitates.

## 3. Results and Discussion

### 3.1. Effect of the Initial pH and Current Density on Struvite Electrochemical Precipitation Performance

#### 3.1.1. Nutrients Removal from Synthetic Wastewater Containing Only Orthophosphate Phosphorus and Ammonium Nitrogen

Twelve batch tests were conducted under controlled initial pH (6.5, 7.0, 7.5 and 8.0) and current density (2, 4 and 6 mA/cm^2^). Due to the similar removal trends, only batch tests with different pHs (6.5, 7.0, 7.5 and 8.0) and a fixed current density of 4 mA/cm^2^ with different current densities (2, 4 and 6 mA/cm^2^) and a fixed initial pH of 7.0 were discussed.

Figure 2 shows the effect of different initial pHs on removing orthophosphate phosphorus and ammonium nitrogen at a fixed current density of 4 mA/cm^2^ and the pH changes during the tests. Solution pH increased with time during tests from 6.5 to 8.7, from 7.0 to 8.8, from 7.5 to 8.8, and from 8.0 to 8.9, at the initial pH of 6.5, 7.0, 7.5, and 8.0, respectively; this was mainly due to the generation of hydroxide ions at the cathode through the electrochemical reduction of water [24]. The concentration of orthophosphate phosphorus and ammonium nitrogen decreased faster at the initial pH of 8.0 than at the initial pH of 6.5, 7.0, and 7.5 during the first 20 min, whereas they decreased to the approximate value after 30 min at all initial pHs. This phenomenon might be ascribed to the conditional solubility product of struvite that first decreased to the minimum as initial pH increased but began to increase as pH continued to move up (≥8.0) during the reaction [25,26]. It might also be related to the significantly decreased saturation index of struvite at low pH [13]. Solution pH affected not only the conditional solubility product of precipitates but also the status of Mg^2+^, NH_4_^+^, and PO_4_^3−^ in the solution [27]. The slowing down of the reaction rate in the last 10 min was mainly due to the lack of the reaction substrate of orthophosphate phosphorus. In addition, the molar ratio of removed orthophosphate phosphorus and ammonium nitrogen during all tests was 1.002–1.021, closed to the theoretical molar ratio of 1 of phosphorus and nitrogen in struvite, indicating struvite as the dominant species of the precipitate.

Figure 3 shows the effects of different current densities (2, 4 and, 6 mA/cm^2^) on removing orthophosphate phosphorus and ammonium nitrogen at a fixed initial pH of 7.0. The decrease in orthophosphate phosphorus and ammonium nitrogen concentrations with time enhanced with increasing the current density from 2 to 4 to 6 mA/cm^2^. The higher the current density, the more Mg^2+^ ions released during the same reaction time, and thus the more orthophosphate phosphorus and ammonium nitrogen were removed from the solution via struvite precipitation.

The kinetics of struvite electrochemical precipitation was further investigated. First- and second-order models did not provide a satisfactory fit to the experimental data with a reasonable R-square value. The kinetic data were approximately fitted to a zero-order kinetic model (C = C_0_ − Kt, where C is the residual concentration of the reactant, C_0_ is the initial concentration of the reactant, k is the rate constant, and t is reaction time). As shown in Figure 3, the values of K for phosphate and ammonium reaction kinetics were 1.8135 and 0.8170 min^−1^ at the current density of 2 mA/cm^2^. They were then increased to 3.2735 and 1.5185 min^−1^ at the current density of 4 mA/cm^2^, and further increased to be 4.6302 and 2.1019 min^−1^ at the current density of 6 mA/cm^2^, indicating that the suitable current density might be 4 mA/cm^2^ since the constant rate showed an increase with a slight slowing trend corresponding to the current density of 6 mA/cm^2^. This phenomenon might be ascribed to the applied current density controlled by the electrochemical reactions (e.g., electrodissolution rate, gas evolution, electroflotation, water reactions, and so on) as well as their kinetics. However, a high current density may negatively affect electrolytic efficiency. More specifically, the anodic surface can be isolated by forming stable oxide layers due to oxidation reactions that promote the corrosion phenomena, generating passivation effects. The passivation of the sacrificial anode increases the ohmic resistance, and consequently, the cell potential rises, increasing the operational costs, but the passivation decreases the electrolytic efficiency considerably [28]. As shown in Figure 4, compared with low current density, the serious surface passivation for forming dense oxide layers occurred at the current density of 6 mA/cm^2^. Therefore, from the economic point of view, the optimal current density of 4 mA/cm^2^ was selected for the subsequent batch tests.

In addition, the orthophosphate phosphorus and ammonium nitrogen concentrations showed a slight decrease in the first 10 min followed by a noticeable drop after 10 min during all tests in Figure 3. During the electrochemical precipitation, the nutrients recovery by struvite crystallization was essentially achieved by adjusting solution pH and the release of Mg^2+^ ions. This trend of residual orthophosphate phosphorus and ammonium nitrogen concentrations could be explained by the concentration change of magnesium, ammonium nitrogen, and orthophosphate phosphorus in the solution with the electrolysis time at a different current density. In reality, the struvite formation was controlled by the practical ionic activity product of struvite in solution. Only when it exceeded the thermodynamic solubility product of struvite, the precipitation occurred [29].

#### 3.1.2. Precipitate Analysis from Synthetic Wastewater Containing Only Orthophosphate Phosphorus and Ammonium Nitrogen

A comprehensive investigation for the precipitates produced from the batch tests was conducted. XRD spectra of precipitates under different combinations of initial pH and current density (shown in Figure 5) showed high similarity in the position of peaks and relative peaks intensities to the spectrum of analytical grade struvite (98%), demonstrating the presence of struvite in precipitates. To further confirm the XRD findings, the obtained precipitates were redissolved to analyse their elemental composition and evaluate the effect of different operational parameters (initial pH and current density) on struvite purity. Figure 6 shows a high purity (>88%) of struvite produced in all precipitates. Struvite purity was moderately dependent on the initial pH and current density. The optimal initial pH for obtaining a precipitate with a struvite content (94.2%) was 7.0 at the current density of 4 mA/cm^2^ with deionized water as solute. The correlation between the purity and current density observed in the present study may be explained by the different magnesium release rates at the different current densities, which indirectly affected the Mg:P molar ratio in the vicinity of the anode [13]. It was reported that when the pH was low (<7), phosphate existed as an acid salt, and Mg(H_2_PO_4_)_2_ was the main product, thus resulting in a relatively low purity at the initial pH of 6.5 in this study. If the pH value of the solution was moderate, MgNH_4_PO_4_·6H_2_O was produced. When the pH value was too high (>10), precipitates of Mg_3_(PO_4_)_2_ and Mg(OH)_2_ were mainly produced, which were less easily dissolved in water [30]. When the initial pH was higher than 7, the purity of the product decreased slightly, probably because of the formation in different degrees of other phosphate-based compounds (Mg(HPO_4_)·3H_2_O and Mg_3_(PO_4_)_2_) with a saturation index close to struvite in the pH range of 6.5 to 9.0 [13].

Based on these observations, the harvested electrochemical precipitate and pure struvite were similar at neutral and slightly alkaline conditions. The superior performance resulting for struvite content at neutral and slightly alkaline conditions was also in accordance with another study concerning the formation of pure struvite at neutral pH by electrochemical deposition [31]. Therefore, considering less chemical for pH adjustment and high purity of produced struvite, the initial pH of 7.0 was selected for the subsequent batch tests.

### 3.2. Effect of Organic Substances on Struvite Electrochemical Precipitation Performance

#### 3.2.1. Nutrients Removal from Synthetic Wastewater Containing Orthophosphate Phosphorus, Ammonium Nitrogen and Organics with Major Ions

Figure 7a–d show the effects of BSA, SA, HA, and their combinations on struvite electrochemical precipitation at the initial pH of 7 and current density of 4 mA/cm^2^, respectively. The control was synthetic wastewater containing orthophosphate phosphorus, ammonium nitrogen and major ions. The decrease in orthophosphate phosphorus and ammonium nitrogen concentrations during the batch test for the control was only 5.7 and 2.2 mg/L less than the abovementioned synthetic wastewater containing only orthophosphate phosphorus and ammonium nitrogen, showing little effect on nutrients removal from the major ions. Due to similar removal trends, the orthophosphate phosphorus removal from the treated solutions was mainly analysed for the following discussion. As observed in Figure 7a–c, the individual presence of BSA and SA had a less adverse effect on nutrients removal than HA. The final orthophosphate phosphorus concentrations were 13, 15, and 17 mg/L at a BSA concentration of 100, 200, and 500 mg/L, and were 15, 18, and 20 mg/L at an SA concentration of 100, 200, and 500 mg/L, and were 36, 47, and 85 mg/L at an HA concentration of 200, 500, and 1500 mg/L. The removed orthophosphate phosphorus concentration that decreased with the increase in the concentration of individual organic substances, showing a positive correlation between the inhibition effects and the concentration of individual organic substances. For details, the inhibition effect of SA was slightly greater than that of BSA on the struvite electrochemical precipitation, while the inhibition effect of HA was more than twice as large as BSA and SA under the same concentration. For BSA, the chelation of Mg^2+^ by protein molecules might adversely affect the growth of the struvite crystal and thus decrease nutrients removal [17]. For SA, the carboxy groups might form a gelatum complex with Mg^2+^ to reduce its amount for the crystallization of struvite as well as envelop the crystal nucleus and inhibit the growth of crystals [17]. For HA, its multifunctional groups (e.g., carboxylic, phenolic, carbonyl, hydroxyl groups) might easily interact with the surface moieties of struvite crystals and promote the occurrence of colloid-like adsorption, ionic exchange, complex formation, and oxidation/reduction [15,16]. As shown in Figure 8, the major forms of phosphate are H_2_PO_4_^−^, HPO_4_^2−^, and PO_4_^3−^ under a pH lower than 8.0, 8.0–12.0, and above 12.0, respectively. At a weak alkaline pH (7.0–8.0), carboxyl groups present in the HA have higher affinity for Mg^2+^ than phosphate in the form of H_2_PO_4_^−^, thus inhibiting the struvite growth [16]. In Figure 7c, when pH > 8, the high addition of 1500 mg/L HA showed a lower inhibitory effect on phosphate removal efficiency; this was mainly attributed to the higher solubility of HA and the lower affinity of the carboxyl group for Mg^2+^ than phosphate in the form of HPO_4_^2−^ and PO_4_^3−^. The other likely reason was that HA was soluble under alkaline conditions but tended to form aggregates in the solution at neutral and weak alkaline pH [32]. The macromolecular aggregates were readily coprecipitated and inhibited crystal growth by blocking the active sites of the newly formed nuclei of struvite. The phenomenon observed during the experiments was that the solution containing HA was turbid (difficult to filter, and the turbidity increased with the increasing concentration of HA) at the beginning of the reaction and became an easier filtration after a period of reaction time helped verify it.

As shown in Figure 7d, the combined effects of HA and SA on nutrients removal was greater in the case of a double combination of organic substances. The final orthophosphate phosphorus concentrations were 36, 72, and 92 mg/L for the addition of 500 mg/L SA + 500 mg/L BSA, 1500 mg/L HA + 500 mg/L BSA and 1500 mg/L HA + 500mg/L SA, respectively. For the addition of 1500 mg/L HA + 500mg/L SA + 500 mg/L BSA, no visible precipitates were observed, and the orthophosphate phosphorus removal efficiency was very low (7.6%), whereas orthophosphate phosphorus removal efficiency reached 74.2% at the low addition of 200 mg/L HA + 100mg/L SA + 100 mg/L BSA. Additionally, the combination of 500 mg/L SA + 500 mg/L BSA showed accelerating inhibition effects, while the combination of 1500 mg/L HA + 500 mg/L BSA and 1500 mg/L HA + 500mg/L SA showed antagonistic inhibition effects. The low concentration combination of BSA, SA, and HA showed antagonistic inhibition effects, while the high concentration combination showed accelerating inhibition effects. Specifically, the final orthophosphate phosphorus concentrations were 17, 20, 85, and 122 mg/L for the addition of 500 mg/L BSA, 500 mg/L SA, 1500 mg/L HA and 500 mg/L BSA + 500 mg/L SA + 1500 mg/L HA, respectively. If the control value of 11 mg/L was subtracted from the abovementioned values, the individual inhibitory effect of BSA, SA, and HA would be equivalent to 6, 9, and 74 mg/L, resulting in a sum of 89 mg/L, whereas the actual value for BSA+SA+HA was 111 mg/L, indicating that the combination of 500 mg/L BSA + 500 mg/L SA + 1500 mg/L HA strengthened the inhibitory effect; this clearly showed that the nutrients removal via struvite electrochemical precipitation was significantly affected by different kinds, concentrations, and combinations of organic substances.

#### 3.2.2. Precipitate Analysis from Synthetic Wastewater Containing Orthophosphate Phosphorus, Ammonium Nitrogen and Organics with Major Ions

XRD, SEM, and purity analysis were performed to identify the features of the precipitates recovered (shown in Figure 9, Figure 10 and Figure 11, respectively). From Figure 9, there were no significant differences in the position of the peaks between analytical grade struvite and the harvested precipitates in the absence or presence of organic substances, indicating that neither the major ions nor organic substances entered the crystal lattice of struvite, thus affecting the crystal structure. From Figure 10, the morphologies of the precipitates in the presence of organic substances were evidently different from the analytical grade struvite, indicating the effect of organic substances on the morphogenesis of struvite. The analytical grade struvite in Figure 10a had a prismatic shape, corroborating those reported by other researchers [16,31]. From Figure 10b, the particles largely retained their prismatic shape for the control. From Figure 10c,d, no significant differences in morphology were seen between the harvested precipitates in the presence of BSA and analytical grade struvite. Although prismatic crystals were the majority, the change from a twinned shape to stubby prism could not be overlooked with the increase in BSA concentration. With the presence of SA in Figure 10e,f, the shape of the crystal varied from a triangular prism to pyramidal frustum as the SA concentration increased from 200 to 500 mg/L. Fine and amorphous crystals were the majority in the presence of HA in Figure 10g,h, and a few brick-shaped crystals were observed in the presence of 200 mg/L HA but disappeared in the presence of 500 mg/L HA. It appeared that HA not only significantly affected the morphology of struvite but also effectively inhibited the crystal growth of struvite. With the presence of various combinations, the produced precipitates were amorphous and coarse in Figure 10i–k and the crystal morphology was a trapezoidal prism in Figure 10l. However, all the particle sizes were irregular with a rough surface, especially for the combinations with a high concentration of HA. The phenomena further proved the blocking of active sites for newly formed nuclei of struvite by the adsorption of organic substances on a crystal surface.

As shown in Figure 11, the decrease of struvite purity in precipitates produced from the control was only 0.3% less than the synthetic wastewater containing only orthophosphate phosphorus and ammonium nitrogen, showing little effect on the quality of precipitates from major ions. The struvite purities and orthophosphate phosphorus removal efficiencies were 92.0%, 91.3%, 89.9%, and 90.2%, 88.6%, 87.1% at a BSA concentration of 100, 200, and 500 mg/L, and were 91.2%, 89.9%, 88.8%, and 88.5%, 86.4%, 84.9% at an SA concentration of 100, 200, and 500 mg/L, and were 85.1%, 83.5%, 75.7%, and 72.7%, 64.4%, 35.6% at an HA concentration of 200, 500, and 1500 mg/L, correspondingly. It clearly shows that the struvite purity decreased with the increase of SA, BSA, and HA concentration; the adverse effect of purity was the greatest for the addition of HA, which also showed a positive correlation with nutrients removal efficiencies. Similar results were obtained for the combined effects of nutrients removal efficiencies, and the combinations of 500 mg/L BSA + 500 mg/L SA, 500 mg/L BSA + 1500 mg/L HA, 500 mg/L SA + 1500 mg/L HA, and 100 mg/L BSA+100 mg/L SA + 200 mg/L HA showed antagonistic inhibition effects for struvite purity.

### 3.3. Preliminary Tests to Mitigate the Adverse Effect of Organic Substances on Struvite Electrochemical Precipitation Performance

From the analysis presented above, nutrients removal and precipitate quality were significantly affected by different kinds, concentrations, and combinations of organic substances, especially the high concentration of HA + BSA + SA. Moreover, there was the phenomenon that all three organic substances and different combinations could inhibit the orthophosphate phosphorus removal rate at the beginning of the reaction at a high concentration but the inhibition was mitigated over time with the increase of pH. Thus, feasibility tests for investigating the change in initial pH to resist high concentration organic substances were conducted. As shown in Figure 12, the high concentration of organic substances with the addition of 500 mg/L BSA, 500 mg/L SA, and 1500 mg/L HA completely suppressed the struvite electrochemical precipitation reaction at the initial pH of 7 and current density of 4 mA/cm^2^. While the initial pH increased to 8 and 9, the orthophosphate phosphorus removal efficiency increased from 7.6% to 53.0% and 56.1% in 50 min and reached 88.6% and 90.2% in 100 min due to the high solubilization of HA under alkaline conditions [32]. In fact, these conditions provided an orthophosphate phosphorus removal of up to 87.1% for an extended period of 40 min at the initial pH of 8. From the economic point of view, the optimal initial pH of 8 was more conducive to nutrients removal from the wastewater with high concentration organic substances (e.g., anaerobically treated swine wastewater by conventional anaerobic digestion). The experimental results demonstrated that an increment in initial pH may mitigate the inhibition of nutrients removal by high concentration organic substances. Struvite electrochemical precipitation may be a promising process for recovering nutrients in the presence of high concentration organic substances at a relatively low pH of 8; this can help improve struvite electrochemical precipitation techniques in practical application.

## 4. Conclusions

In this study, laboratory-scale batch tests were conducted to investigate the effects of initial pH, current density, and organic substances (BSA, SA, and HA) from synthetic anaerobically treated swine wastewater on struvite electrochemical precipitation performance. Under the optimal conditions of the initial pH of 7.0 and current density of 4 mA/cm^2^ in the absence of organic substances, the orthophosphate phosphorus and ammonium nitrogen removal efficiencies and purity of precipitates produced were 96.2%, 11.5% and 94.2%, respectively. The nutrients removal and precipitate quality were significantly affected by different kinds, concentrations, and combinations of organic substances. XRD analysis revealed that the crystal form of the struvite precipitates produced was not changed in the presence of organic substances. SEM analysis revealed that compared with BSA, SA, and HA, the morphology of struvite crystals significantly changed. With the increase in concentration from 200 to 500 mg/L, the crystal morphology varied from a twinned shape to stubby prism, from triangular prism to pyramidal frustum, and from a few brick-shaped crystals to fine and amorphous crystals, in the individual presence of BSA, SA, and HA, respectively. While in the presence of various combinations, the precipitates produced were amorphous, and all the particles were irregular with a rough surface, except for trapezoidal prism shapes in the low concentration combination. Purity analysis revealed that there was a positive correlation with nutrients removal efficiencies. Consequently, more adverse effects were revealed upon nutrients removal, morphology, and purity of precipitates by HA than BSA and SA. High concentration organic substances with the addition of 500 mg/L BSA, 500 mg/L SA, and 1500 mg/L HA completely suppressed the struvite electrochemical precipitation reaction (inhibition ratio of 92.4%) at the initial pH of 7.0 and current density of 4 mA/cm^2^, whereas this inhibition effect was mitigated to 13% by adjusting initial pH to 8.0 for an extended period of 40 min; this may help improve struvite electrochemical precipitation techniques in practical application for recovering nutrients from anaerobically treated swine wastewater.

## Figures and Tables

**Figure 1 membranes-11-00594-f001:**
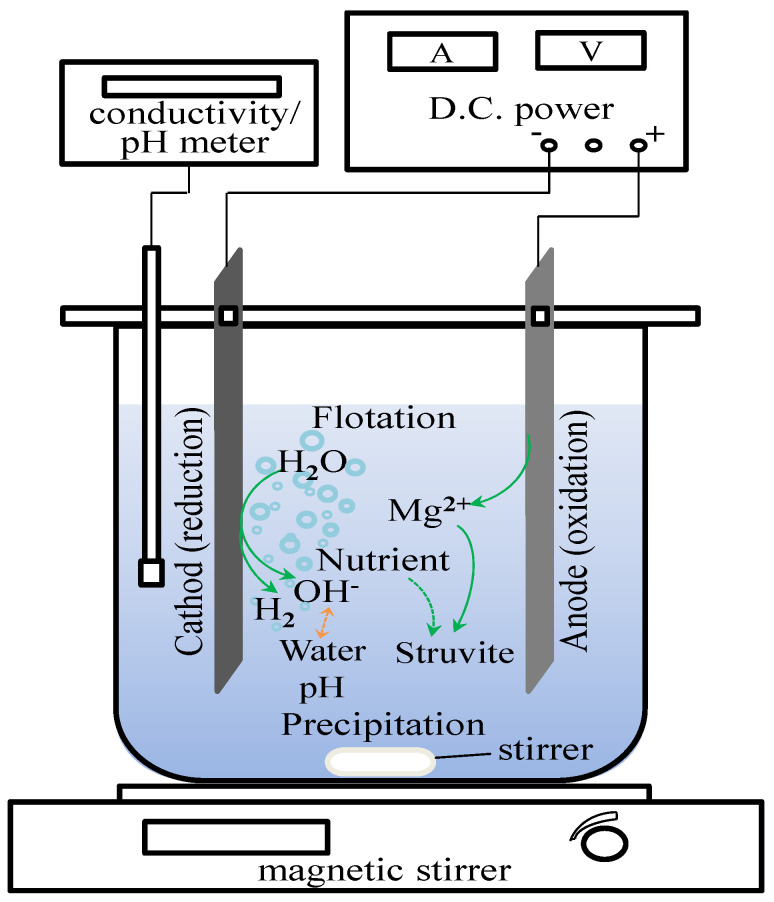
Schematic diagram of experimental set-up.

**Figure 2 membranes-11-00594-f002:**
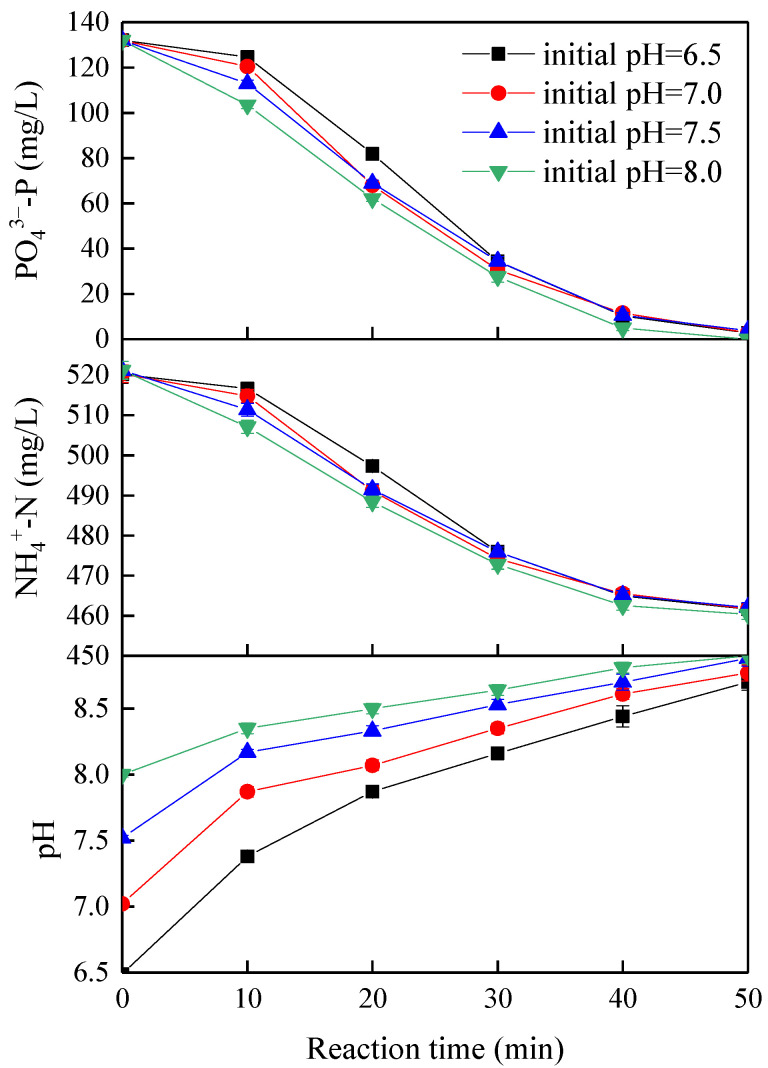
Effect of initial pH on nutrients removal (current density of 4 mA/cm^2^).

**Figure 3 membranes-11-00594-f003:**
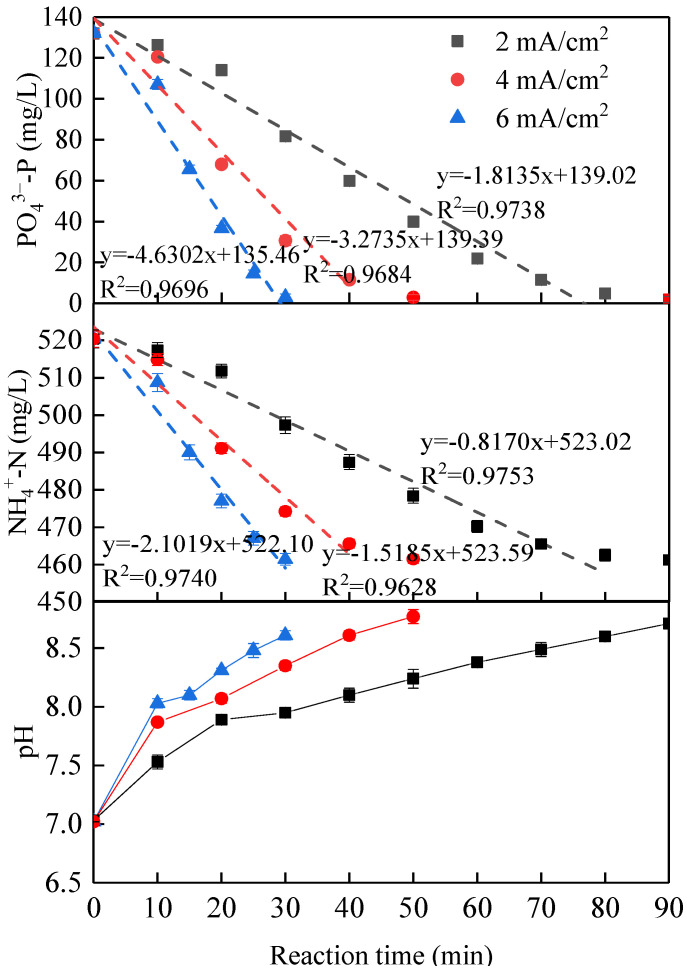
Effect of the current density on nutrients removal (initial pH of 7.0).

**Figure 4 membranes-11-00594-f004:**
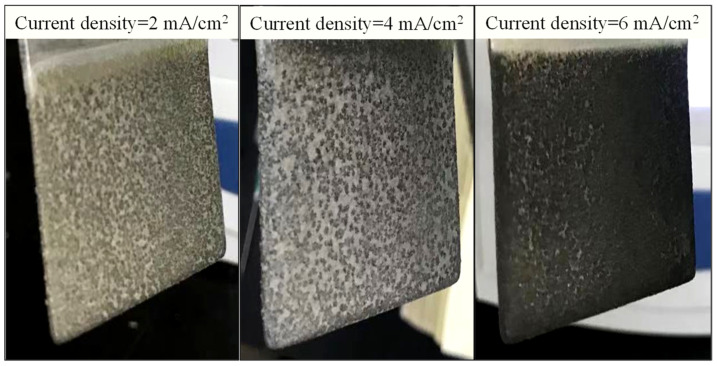
Visual observation of Mg anode at various current densities and the same pH of 7.0.

**Figure 5 membranes-11-00594-f005:**
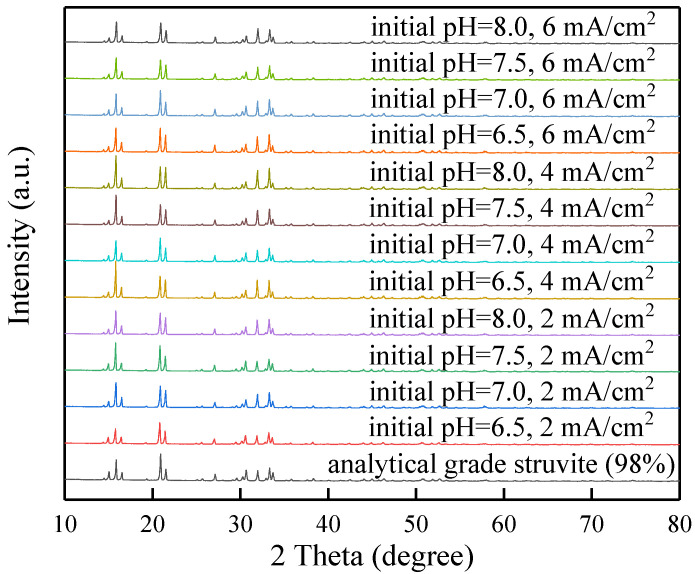
XRD diffractogram of precipitates.

**Figure 6 membranes-11-00594-f006:**
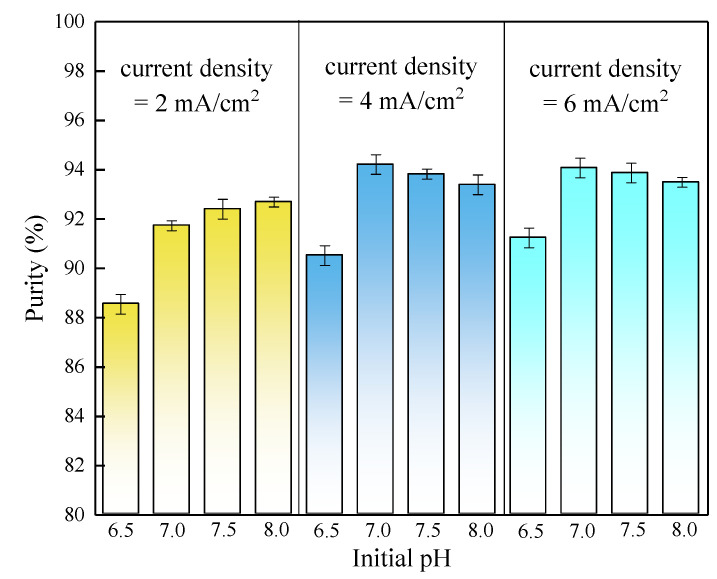
Struvite purity in the precipitates.

**Figure 7 membranes-11-00594-f007:**
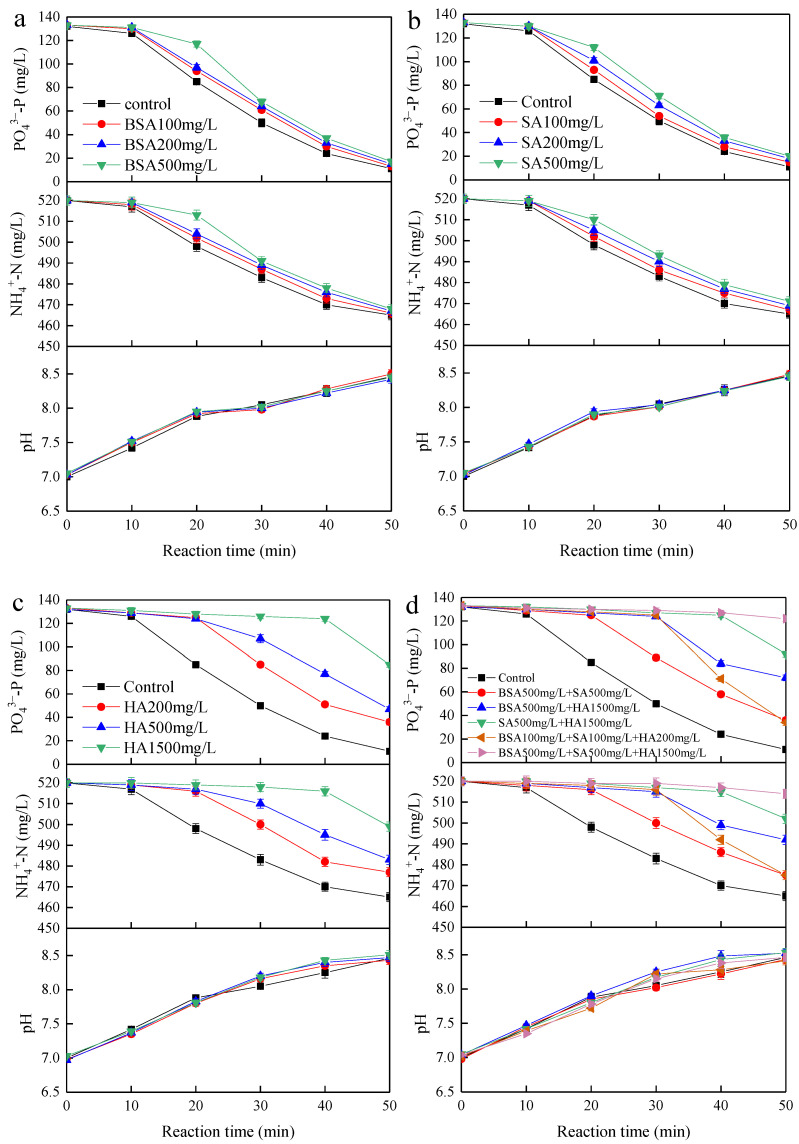
Effect of organic substances under different concentrations: (**a**) for BSA, (**b**) for SA, (**c**) for HA and (**d**) different combinations on nutrients removal.

**Figure 8 membranes-11-00594-f008:**
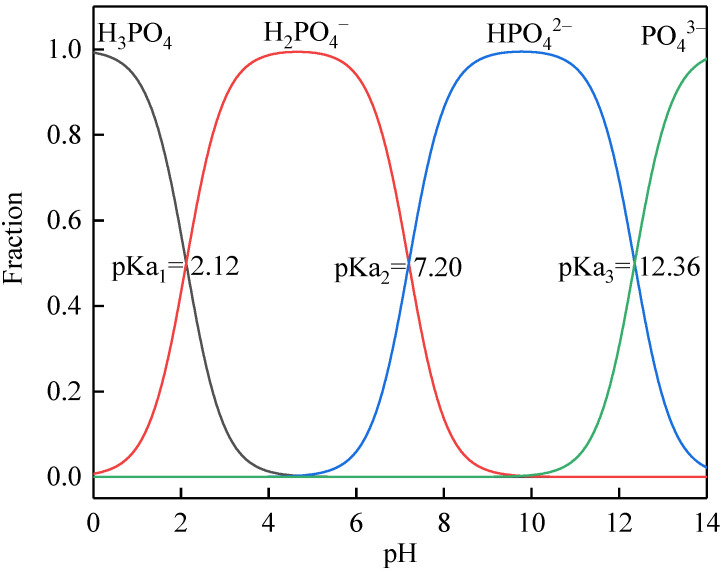
Distribution of phosphate species as a function of pH.

**Figure 9 membranes-11-00594-f009:**
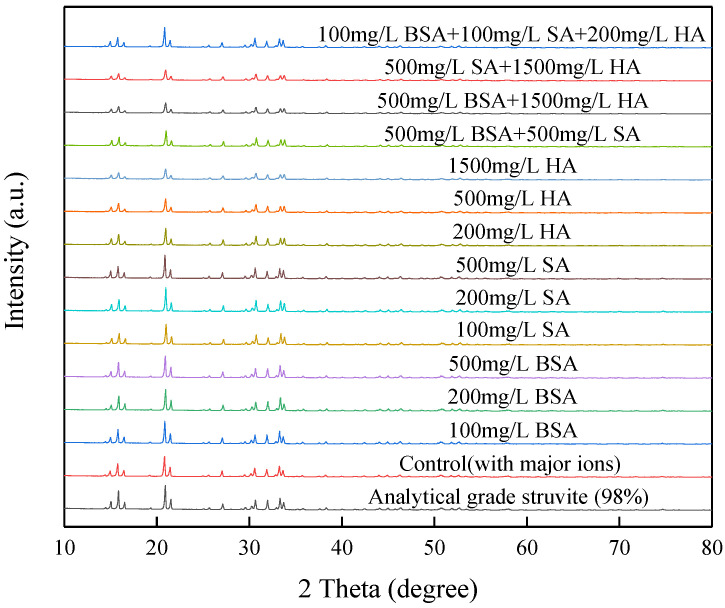
XRD diffractogram of precipitates.

**Figure 10 membranes-11-00594-f010:**
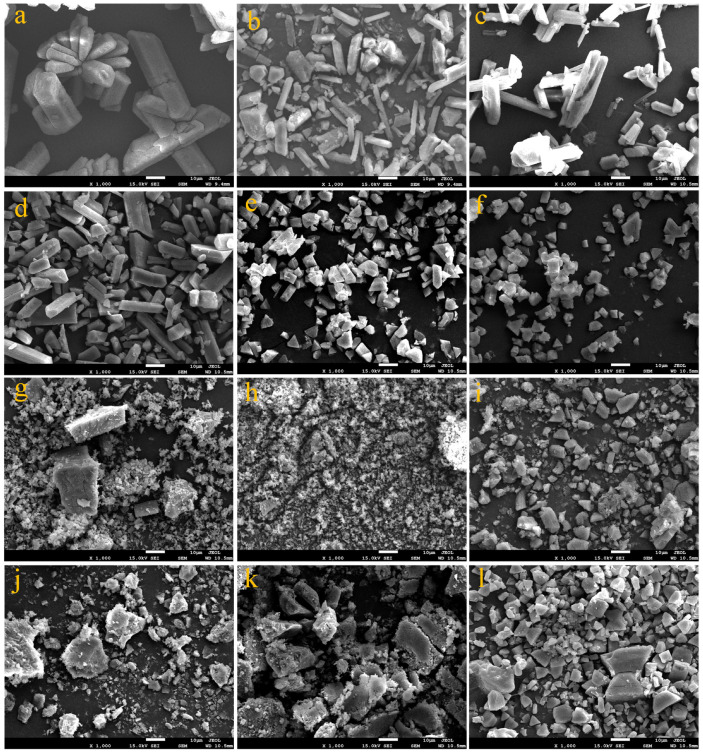
SEM images of precipitates for (**a**) analytical grade struvite (98%); (**b**) control; at different concentrations: (**c**) 200 mg/L BSA, (**d**) 500 mg/L BSA, (**e**) 200 mg/L SA, (**f**) 500 mg/L SA, (**g**) 200 mg/L HA, (**h**) 500 mg/L HA; at different combinations: (**i**) 500 mg/L BSA + 500 mg/L SA, (**j**) 500 mg/L BSA + 1500 mg/L HA, (**k**) 500 mg/L SA + 1500 mg/L HA, (**l**) 100 mg/L BSA + 100 mg/L SA + 200 mg/L HA.

**Figure 11 membranes-11-00594-f011:**
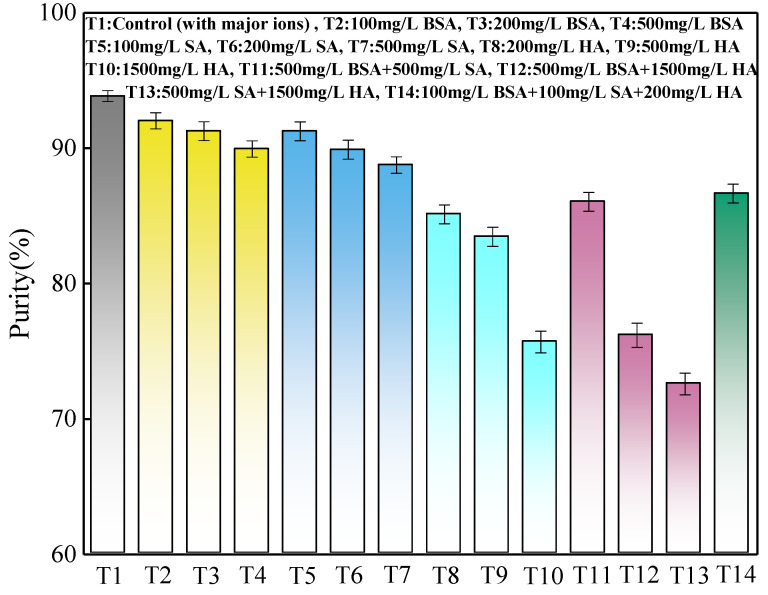
Struvite purity in the precipitates.

**Figure 12 membranes-11-00594-f012:**
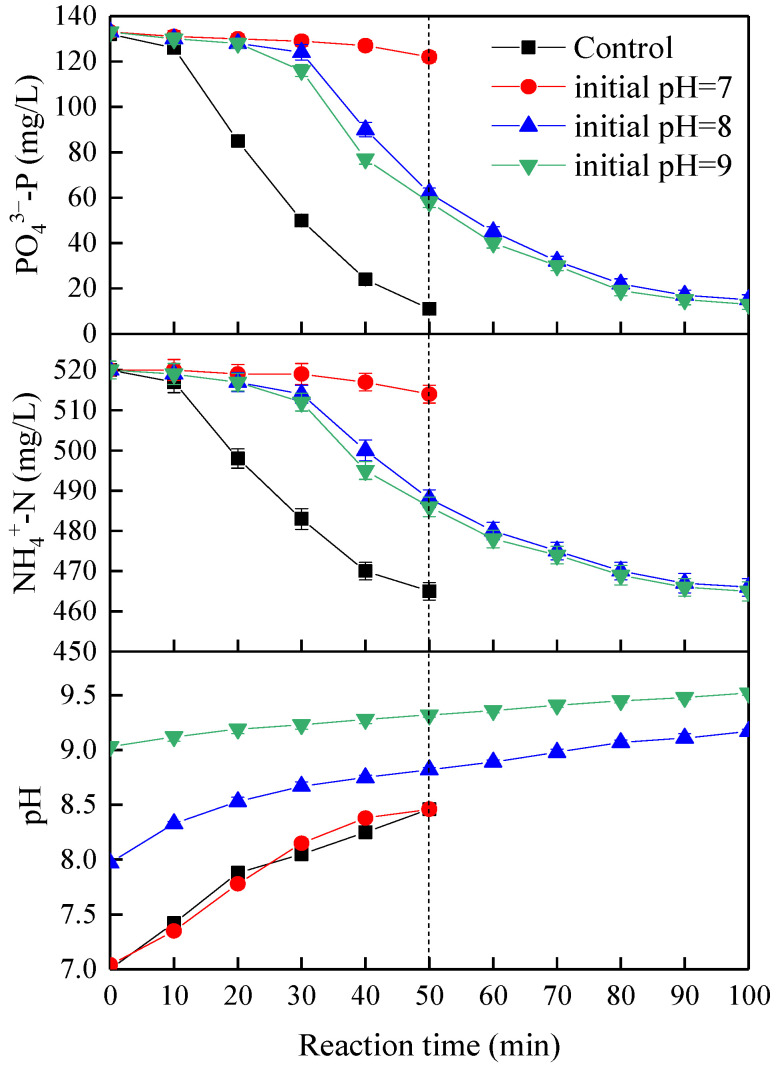
Feasibility tests for mitigating the effect of high concentration organic substances by adjusting initial pH at a current density of 4 mA/cm^2^.

## Data Availability

Not applicable.

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
