# Peer review of "Effect of Organic Substances on Nutrients Recovery by Struvite Electrochemical Precipitation from Synthetic Anaerobically Treated Swine Wastewater"

_membranes, 2021, doi:10.3390/membranes11080594_

Round 1

Reviewer 1 Report

The article is interesting considering the subject and the research methods. My romandations refers to the future studies of the authors, to extend the present research to other type of wastewater (real wastewater), other experimental parameters...

Author Response

Thanks for your comment. As you suggested, we are extending the present research to the real swine wastewater and comparing the performance of struvite electrochemical precipitation for anaerobically treated swine wastewater by anaerobic sequencing batch reactor and anaerobic membrane bioreactor, respectively. The effects of major experimental parameters on struvite electrochemical precipitation performance, such as initial pH, current density and the molar ratio of NH4+ : PO43-, will be also considered for treating real wastewater.

Reviewer 2 Report

The manuscript deals with "Effects of Organic Substances on Nutrients Recovery by Struvite Electrochemical Precipitation from wastewater". The manuscript has been written well. There is just a typo in the manuscript as below:

1. Page 1, Line 15; "to investigate the effects of initial pH, current density and organic substances from synthetic wastewater on nutrients removal and precipitates quality".

Delete the "from synthetic wastewater"

Author Response

Thanks for your comment. We deleted the "from synthetic wastewater" in Line 16 in the revised manuscript.

Reviewer 3 Report

The topic of your submitted paper is interesting and some valuable results were obtained.  There are some unclear points in your submitted paper. I have judged major revision is required before accepting your paper. Please revise your paper by taking into accounts the following points.

  1. The way of writing is too redundant. It took long time to read your paper. Please write your paper briefly. Especially, you inserted many precise concentrations in the text. This way of writing makes your paper redundant and difficult to read.
  2. I could not understand the composition of synthetic anaerobically treated swine wastewater. Anaerobically treated swine wastewater normally contains high concentration of suspended solid. This high SS concentration make difficult to treat swine wastewater. You must consider this point in your paper. Also you did not consider the cation and trace element concentration for synthetic anaerobically treated swine wastewater.
  3. You must use orthophosphate phosphorus instead of phosphate phosphorus.
  4. It is better not to use “the” in the title of Figure.

Author Response

The topic of your submitted paper is interesting and some valuable results were obtained. There are some unclear points in your submitted paper. I have judged major revision is required before accepting your paper. Please revise your paper by taking into accounts the following points.

Comment 1:The way of writing is too redundant. It took long time to read your paper. Please write your paper briefly. Especially, you inserted many precise concentrations in the text. This way of writing makes your paper redundant and difficult to read.

Re: Thanks for your comment. We mainly rewrote the redundant and repeated parts of inserting many precise concentrations instead of brief description in Line 447-454 in the revised manuscript. For accurate analysis, other precise concentrations were required and retained in the article. Hope you agree with it.

Comment 2:I could not understand the composition of synthetic anaerobically treated swine wastewater. Anaerobically treated swine wastewater normally contains high concentration of suspended solid. This high SS concentration make difficult to treat swine wastewater. You must consider this point in your paper. Also you did not consider the cation and trace element concentration for synthetic anaerobically treated swine wastewater.

Re: Thanks for your comment. We totally agree with your opinion that high concentration (normally hundreds to few tens thousands mg/L) suspended solids in anaerobically treated swine wastewater by conventional processes (e.g., anaerobic digestion, anaerobic sequencing batch reactor, anaerobic baffled reactor) would deteriorate the performance of struvite electrochemical precipitation and the quality of produced struvite precipitates, making it a huge challenge for struvite electrochemical precipitation to be used for post-treatment of swine wastewater treated by conventional anaerobic processes. Therefore, we propose the advanced process of anaerobic membrane bioreactor, where the micro/ultra-filtration membrane is employed to replace the gravitational sedimentation for sludge and water separation, to produce the effluent free of suspended solids. Based on this hypothesis, we prepared the synthetic wastewater free of suspended solids to simulate the effluent of anaerobic membrane bioreactor treating swine wastewater in this study. We also conducted the cation and anion scanning via Inductively Coupled Plasma Mass Spectrometry (ICP-MS) and Ion Chromatography (IC), respectively, for a real swine wastewater sample. Besides NH4+ and PO43-, other major ions (K+, CO32- and SO42-) in the synthetic anaerobically treated swine wastewater were also kept as the same with the real swine wastewater in order to cover their potential effects on struvite electrochemical precipitation. Trace elements (including Fe, Zn, Cu, Cr, Co, Ni) were detected with concentration less than 0.2 mg/L in the real swine wastewater. Previous studies have demonstrated their marginal effects on struvite electrochemical precipitation. Thus, we didn't take trace elements into account in this study. We clarified it in Line 48-54, 114-119 in the revised manuscript.

Comment 3:You must use orthophosphate phosphorus instead of phosphate phosphorus.

Re: Thanks for your comment. We corrected it throughout the article in the revised manuscript.

Comment 4:It is better not to use “the” in the title of Figure.

Re: Thanks for your comment. We removed the “the” in the titles of Figure 4, 5, 6, 9 and 11 in the revised manuscript.

Round 2

Reviewer 3 Report

You have revised your submitted paper properly by taking into accounts reviewer’s comments. I have judged your revised paper reached to the acceptance level of Membrane.